# MaskMA: Towards Zero-Shot Multi-Agent Decision Making with Mask-Based Collaborative Learning

**Jie Liu**[*1,3]**, Yinmin Zhang**[*2,3]**, Chuming Li**[2,3]**, Zhiyuan You**[1]**, Zhanhui Zhou**[3]**, Chao Yang**[3]**, Yaodong Yang**[†4]**,
Yu Liu**[5]**, Wanli Ouyang**[1,3]

[*] *Equal contribution,* [†] *Corresponding author. Contact: jieliu@link.cuhk.edu.hk, yaodong.yang@pku.edu.cn*

[1] *Multimedia Laboratory, The Chinese University of Hong Kong.*

[2] *The University of Sydney.*

[3] *Shanghai Artificial Intelligence Laboratory.*

[4] *Institute for AI, Peking University.*

[5] *SenseTime Research.*

*Reviewed on OpenReview:* *https://openreview.net/forum?id=Susy8EAff9*

## Abstract

Building a single generalist agent with strong zero-shot capability has recently sparked significant advancements. However, extending this capability to multi-agent decision making scenarios presents challenges. Most current works struggle with zero-shot transfer, due to two challenges particular to the multi-agent settings: (a) a mismatch between centralized training and decentralized execution; and (b) difficulties in creating generalizable representations across diverse tasks due to varying agent numbers and action spaces. To overcome these challenges, we propose a **Mask**-Based collaborative learning framework for **M**ulti-**A**gent decision making (MaskMA). Firstly, we randomly mask part of the units and collaboratively learn the policies of unmasked units to handle the mismatch. In addition, MaskMA integrates a generalizable action representation by dividing the action space into intrinsic actions solely related to the unit itself and interactive actions involving interactions with other units. This flexibility allows MaskMA to tackle tasks with varying agent numbers and thus different action spaces. Extensive experiments in SMAC reveal MaskMA, with a single model trained on 11 training maps, can achieve an impressive 77.8% average zero-shot win rate on 60 unseen test maps by decentralized execution, while also performing effectively on other types of downstream tasks (*e.g.,* varied policies collaboration, ally malfunction, and ad hoc team play).

## 1 Introduction

The powerful transformer-based (Vaswani et al., 2017) generalist models (Ouyang et al., 2022; Touvron et al., 2023; Brown et al., 2020; Ramesh et al., 2022) have brought artificial intelligence techniques to the daily life of people. The reinforcement learning community has also witnessed a growing amount of successes in transformer-based models (Chen et al., 2021; Carroll et al., 2022; Liu et al., 2022; Janner et al., 2022; Meng et al., 2023). However, most of these existing generalist models either fail to complete unseen tasks effectively or fail to incorporate the multi-agent reality of decision-making. A natural follow-up question is: *how can one build a generalist model for multi-agent decision-making that is capable of zero-shot transfer to new tasks* (*e.g.,* different maps and varying numbers of agents in SMAC (Samvelyan et al., 2019)).

Compared to single-agent scenarios, directly utilizing transformers for centralized training in multi-agent settings encounters two primary challenges. (a) *A mismatch between centralized training and decentralized execution*. Multi-agent decision-making typically follows centralized training with a decentralized execution (Gronauer & Diepold, 2022) approach. Most existing methods (Wen et al., 2022b; Tseng et al., 2022b) treat multi-agent decision-making as a sequence modeling problem and directly employ transformer architectures. However, the centralized training phase of transformers utilizing all units as inputs mismatches with the decentralized execution phase where each agent's perception is limited to only nearby units. This mismatch significantly reduces performance. (b) *Generalization*

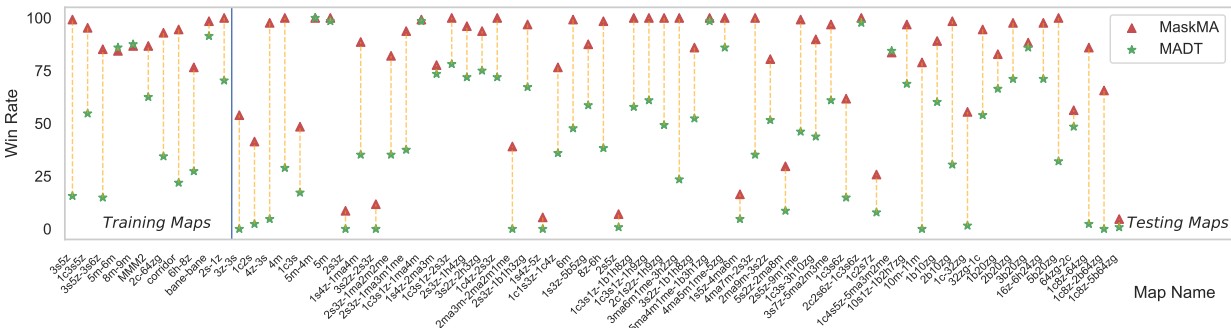

Figure 1: **Win rate on training and testing maps.** The blue line separates the 11 training maps on the left from the 60 testing maps on the right, where the performance on the testing maps is the zero-shot performance. The orange line demonstrates the substantial performance advantage of MaskMA over MADT (Multi-Agent Decision Transformer (Meng et al., 2023)).

*problems caused by varying numbers of agents and actions.* Downstream tasks have different numbers of agents, resulting in varying action spaces. Meng et al. (2023) takes a large action space and mutes the unavailable actions using an action mask to handle the variable action spaces. However, this method suffers from poor generalization because the same component of the action vector represents different physical meanings in tasks with different numbers of agents.

To address these challenges, we ~~propose~~ introduce two scalable techniques: a Mask-based Training Strategy (MTS) and a Generalizable Action Representation (GAR). The two techniques form the basis of a new mask-based collaborative learning framework for multi-agent decision-making, named MaskMA. To address the mismatch challenge, we incorporate random masking into the attention matrix of the transformer, effectively reconciling the discrepancy between centralized training and partial observations in decentralized execution, thus bolstering the model's generalization capabilities. To handle the generalization challenge, MaskMA integrates GAR by categorizing actions into two types: intrinsic actions, which are solely related to the unit itself, and interactive actions, which involve interactions with other units. GAR predicts one unit's interactive action from its own features together with other units' features, instead of only relying on its own features, allowing MaskMA to generalize to diverse tasks with varying agent numbers and action spaces.

We evaluate MaskMA's performance using the StarCraft Multi-Agent Challenge (SMAC) benchmark. To validate the zero-shot performance, we evaluate the win rate in a challenging setting, using only 11 maps for training and 60 unseen maps for testing. Extensive experiments demonstrate that our model significantly outperforms the previous state-of-the-art methods in zero-shot scenarios. We also introduce various downstream tasks to further verify the ~~strong~~ robust generalization ability of MaskMA, including varied policies collaboration, ally malfunction, and ad hoc team play. MaskMA is the first approach that achieves ~~strong~~ notable zero-shot capability for multi-agent decision-making (*e.g.,* 78% win rate in SMAC). We hope this work lays the groundwork for further advancements in multi-agent fundamental models, with potential applications across a wide range of domains. Our main contributions are summarized as three folds:

- We introduce a novel mask-based collaborative learning framework, MaskMA, for multi-agent decision-making. This framework pretrains a transformer architecture with a Mask-based Training Strategy (MTS) and a Generalizable Action Representation (GAR).

- We set up a challenging zero-shot setting for general models in multi-agent scenarios based on the SMAC, *i.e.,* training on only 11 maps and testing on 60 different maps and three downstream tasks including varied policies collaboration, ally malfunction, and ad hoc team play.

- To our best knowledge, MaskMA is the *first* general model for multi-agent decision-making with notable zero-shot performance. With only 11 training maps, our MaskMA has achieved an impressive average zero-shot win rate of 77.8% on 60 unseen test maps, representing a 78% improvement relative to the baseline method.

## 2   Related Work

**Masked Training in Single-agent Decision Making.**   DT (Chen et al., 2021) casts the reinforcement learning as a sequence modeling problem conditioned on return-to-go, using a transformer to generate optimal action. MaskDP (Liu et al., 2022) utilizes autoencoders on state-action trajectories, learning the environment's dynamics by masking and reconstructing states and actions. Uni[MASK] (Carroll et al., 2022) expresses various tasks as distinct masking schemes in sequence modeling, using a single model trained with randomly sampled maskings. In this paper, we explore the design of sequences in multi-agent decision making and how it can be made compatible with the mask-based training strategy.

**Multi-agent Decision Making as Sequence Modeling.**   MADT (Meng et al., 2023) introduces Decision Transformer (Chen et al., 2021) into multi-agent reinforcement learning, significantly improving sample efficiency and achieving strong performance in SMAC. MAT (Wen et al., 2022a) leverages an encoder-decoder architecture, incorporating the multi-agent advantage decomposition theorem to reduce the joint policy search problem into a sequential decision-making process. Tseng et al. (2022a) utilize the Transformer architecture and propose a method that identifies and recombines optimal behaviors through a teacher policy. ODIS (Zhang et al., 2023) trains a state encoder and an action decoder to extract task-invariant coordination skills from offline multi-task data. In contrast, our proposed MaskMA adapts the Transformer architecture to multi-agent decision making by designing a sequence of inputs and outputs for a generalizable action representation. This approach offers broad generalizability across varying actions and various downstream tasks.

**Action Representation.**   Recent works have explored semantic action in decision-making, especially in multi-agent environments. ASN (Wang et al., 2020) focuses on modeling the effects of actions by encoding the semantics of actions to understand the consequences of agent actions and improve coordination among agents. UPDeT (Hu et al., 2021) employs a policy decoupling mechanism that separates the learning of local policies for individual agents from the coordination among agents using transformers. Yang et al. (2023) proposes a graph network to handle variable action spaces, applying reinforcement learning to improve finite element (Brenner, 2008) methods. Our MaskMA emphasizes sequence modeling, masking strategies, and generalizable action representation, which is similar to Yang et al. (2023), to train a generalist model in the multi-agent field. This model can be applied to unseen maps and a wide range of downstream tasks, involving varied policy collaboration, ally malfunction, and ad hoc team play.

## 3   Method

In this section, we introduce our proposed mask-based collaborative learning framework. We start by providing a formulation of the multi-agent decision-making problem. Then, we introduce the Mask-based Training Strategy (MTS) and the Generalizable Action Representation (GAR).

### 3.1   Formulation

#### 3.1.1   Multi-Agent Preliminaries

In multi-agent scenarios, states are only partially observable, *i.e.*, each agent has only limited sight and can only observe part of other units (*e.g.*, enemies). Therefore, a cooperative multi-agent task is generally defined as a decentralized partially observable Markov decision process (Oliehoek & Amato, 2015), denoted as $G = < \boldsymbol{S}, U, \boldsymbol{A}, P, O, r, \gamma >$. Here $\boldsymbol{S}$ is the global state space, and $U \triangleq \{u_i\}_{i=1}^N$ is the set of $N$ units, where the first $M$ units are controllable and the rest $N - M$ units are uncontrollable, often subsumed as part of the environment. $\boldsymbol{A} = \prod_{i=1}^M A_i$ is the action space for controllable units. At time step $t$, each agent $u_i \in \{u_i\}_{i=1}^M$ selects an action $a_i \in A_i$, forming a joint action $\boldsymbol{a} \in \boldsymbol{A}$. The joint action $\boldsymbol{a}$ at state $\boldsymbol{s} \in \boldsymbol{S}$ triggers a transition of $G$, subject to the transition function $P(\boldsymbol{s'} \mid \boldsymbol{s}, \boldsymbol{a}) : \boldsymbol{S} \times \boldsymbol{A} \times \boldsymbol{S} \to [0, 1]$. Meanwhile, a shared reward function $r(\boldsymbol{s}, \boldsymbol{a}) : \boldsymbol{S} \times \boldsymbol{A} \to \mathbb{R}$ is employed, with $\gamma \in [0, 1]$ denoting the discount factor.

The global state at the $t$-th time step is defined as $\boldsymbol{s}^t = (s_1^t, s_2^t, ..., s_N^t)$, where each $s_i^t$ exclusively represents the state of unit $u_i$, and does not include any information about other units. The local observation $o_i^t$ for each unit $u_i$ is composed of the states of all units visible to it at the $t$-th step, expressed as $o_i^t = \{s_i^t \mid i \in p_i^t\}$, where $p_i^t$ indicates the indexes of

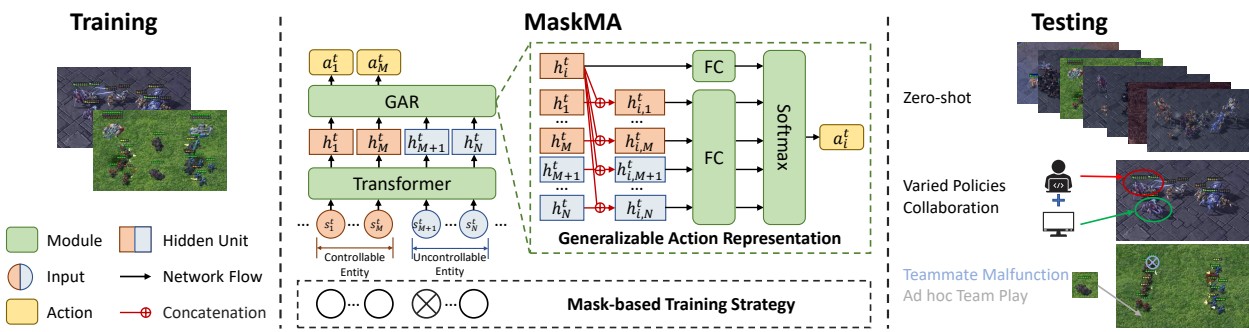

Figure 2: MaskMA employs the transformer architecture combined with generalizable action representation and then trained through a mask-based training strategy. It effectively generalizes skills and knowledge from training maps into various downstream tasks, including unseen maps, varied policies collaboration, ally malfunction, and ad hoc team play.

units within $u_i$'s observation range. Actions are categorized into two types: intrinsic actions, which are solely related to the unit itself, and interactive actions, which involve interactions with other units.

### 3.1.2 Multi-Task Imitation Learning

Since agents have to learn from a broad range of tasks before they can show good transfer to new tasks, we formulate the problem as multi-task imitation learning. Formally, we assume access to a dataset of passively logged expert trajectories $\mathcal{D} = \{\tau_i = \{(\boldsymbol{s}_t, \boldsymbol{a}_t)\}_{t=1}^T\}$ from some training tasks: $\mathcal{T} \sim p(\mathcal{T})$, where $p(\mathcal{T})$ is the task distribution. After imitating the expert demonstrations through supervised learning from $\mathcal{D}$, we then test the agents' performance on new tasks drawn from $p(\mathcal{T})$. However, vanilla-supervised learning fails to produce good transfer in the multi-agent settings due to the mismatch between centralized pretraining and decentralized execution as well as the fact that the varying number of agents may lead to completely new observation and action space that agents have never seen before. Then, we present our strategy to solve these issues.

### 3.2 Mask-Based Training Strategy

To handle the mismatch between centralized training and decentralized execution, we propose to randomly mask some parts of the units in the environment and collaboratively learn the execution policies of other unmasked units.

We utilize a standard causal transformer with only encoder layers as our model backbone, with the context length equal to $L$. At the $t'$-th timestep, the input comprises the recent $L$ global states of $N$ agents, expressed as $(\boldsymbol{s}^{t'-L+1}, \cdots, \boldsymbol{s}^{t'})$, resulting in total $L \times N$ tokens, including all agents in the entire sequence, both controllable and uncontrollable. Then we harness the capabilities of the attention matrix to realize mask-based collaborative learning. The shape of these mask matrices is $(LN \times LN)$, corresponding to $L \times N$ input tokens. During the training phase, we meticulously craft the attention matrix $m_1$ to exhibit a causal structure along the timestep dimension, while maintaining a non-causal configuration within each discrete timestep. This design is illustrated in the left part of Figure 3. Based on a predetermined mask ratio, we then generate the final attention matrix $m_2$ for training (as depicted in the right part of Figure 3) through a process of random sampling, wherein specific elements are set to zero.

For evaluation, we can efficiently shift between centralized and decentralized execution by adjusting the attention mask matrix. For decentralized execution, we modify the attention matrix to ensure that each agent focuses solely on the surrounding agents during the self-attention process. In contrast, for centralized execution, we simply set the attention matrix to be equivalent to $m_1$.

### 3.3 Generalizable Action Representation

As agents involve interactive actions among other units in most multi-agent tasks (*e.g.*, the healing allies and attacking enemies in SMAC), their action space grows with the number of units. To address the challenge posed by variable numbers of agents, we introduce the concept of Generalizable Action Representation (GAR), which is similar to Yang et al. (2023). Figure 2 illustrates the process through which the final action $a_i^t$ of a specific agent is determined by GAR.

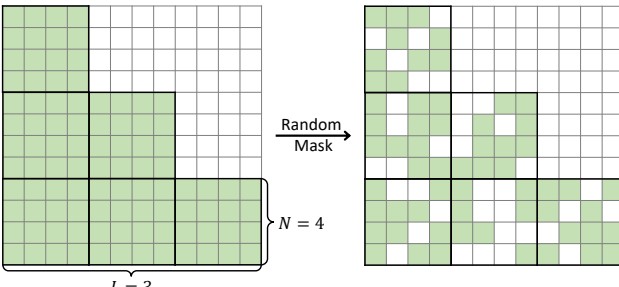

Figure 3: **Visualization of the attention matrix in MTS.** Left: The attention matrix displays a causal structure along the timestep dimension, complemented by a non-causal configuration within each discrete timestep. Right: The final attention matrix used for training is obtained by randomly masking elements of the left attention matrix.

Consider an interaction between the executor unit $u_i$ and the receiver unit $u_j$, we define the embedding of this interaction as $h_{i,j}^t = h_i^t \oplus h_j^t$, where $h_i^t$ and $h_j^t$ are the output embedding of $u_i$ and $u_j$ from the transformer encoder respectively, and $\oplus$ denotes the concatenation operation. For interactive actions, we compute the logits as $\left\{ FC(h_{i,j}^t) \right\}_{j=1}^{N}$, where the fully connected layer $FC$ has an output shape of one. In contrast, the logits for intrinsic actions are obtained using $FC(h_i^t)$, with the output shape of $FC$ corresponding to the number of intrinsic actions. These logits, encompassing both interactive and intrinsic actions, are then combined and input into a softmax function to determine the final action. Please refer to the pseudocode in the Appendix A.4 for the practical implementation.

### 3.4 Loss Function

With Mask-Based Training Strategy (MTS) and Generalizable Action Representation (GAR), our objective is to minimize multi-agent supervised learning loss. This objective is calculated using the cross-entropy loss between predicted actions and the ground truth actions. Our loss function for an action sequence is defined as:

$$\mathcal{L}_{\text{CE}} = -\frac{1}{L}\frac{1}{M}\sum_{i=1}^{L}\sum_{j=1}^{M}\text{CE}(\mathcal{A}_j^i, \mathcal{P}_j^i),$$

where $\text{CE}(a, p)$ is the cross-entropy between the ground truth action and the predicted action probability distribution. Here, $M$ is the number of controllable agents, and $L$ represents the context length. Note that for all agents in an entire input $L \times N$, we only consider the loss for the $M$ controllable agents.

## 4 Experiments

In this section, we design experiments to evaluate the following features of MaskMA. (1) Zero-shot Ability. We conduct experiments on SMAC using only 11 maps for training and up to 60 maps for testing, assessing the model's ability to generalize to unseen scenarios. In SMAC tasks, agents must adeptly execute a set of skills such as alternating fire, kiting, focus fire, and positioning to secure victory. These attributes make zero-shot transfer profoundly challenging. (2) Effectiveness of MTS and GAR for different multi-agent tasks. We conduct ablation studies to figure out (3) Generalization of MaskMA to downstream tasks. We evaluate the model's performance on various downstream tasks, such as varied policies collaboration, ally malfunction, and ad hoc team play. This helps us understand how the learned skills and strategies can be effectively adapted to different situations.

**Setup.** In SMAC (Samvelyan et al., 2019), players control ally units in StarCraft using cooperative micro-tricks to defeat enemy units with built-in rules, and the state of unit $s_i$ includes unit type, position, health, shield, and so on. Our approach differs from existing methods that only consider grouped scenarios, such as Easy, Hard, and Super-Hard maps. Instead, we extend the multi-agent decision-making tasks by combining different units with varying numbers. We include three races: Protoss (colossus, zealot, stalker), Terran (marauder, marine, and medivac), and Zerg (baneling, zergling, and hydralisk). Note that since StarCraft II does not allow units from different races to be on the same team, we have designed our experiments within this constraint. Firstly, we collect expert trajectories as offline datasets from

Table 1: **Win rate on training maps.** Offline datasets consist of 10k or 50k expert trajectories per map collected by specific expert policies. With the mask-based training strategy, MaskMA consistently exhibits high performance in both Centralized Execution and Decentralized Execution. Its generalizable action representation enables seamless adaptation and convergence on maps with diverse characteristics. In contrast, MADT struggles with different action spaces, achieving a win rate of only 51.78% even after extensive training.

| Map Name | # Episode | Return Distribution | Centralized Execution | Decentralized Execution | |
| --- | --- | --- | --- | --- | --- |
| | | | MaskMA (Ours) | MADT | MaskMA (Ours) |
| 3s_vs_5z | 50k | $19.40 \pm 1.89$ | $85.94 \pm 3.49$ | $73.44 \pm 3.49$ | $82.81 \pm 7.81$ |
| 3s5z | 10k | $18.83 \pm 2.48$ | $98.44 \pm 1.56$ | $15.62 \pm 6.99$ | $99.22 \pm 1.35$ |
| 1c3s5z | 10k | $19.51 \pm 1.40$ | $94.53 \pm 4.06$ | $54.69 \pm 8.41$ | $95.31 \pm 1.56$ |
| 3s5z_vs_3s6z | 10k | $19.69 \pm 1.27$ | $85.94 \pm 6.44$ | $14.84 \pm 9.97$ | $85.16 \pm 5.58$ |
| 5m_vs_6m | 10k | $18.37 \pm 3.69$ | $86.72 \pm 1.35$ | $85.94 \pm 5.18$ | $84.38 \pm 4.94$ |
| 8m_vs_9m | 10k | $19.12 \pm 2.57$ | $88.28 \pm 6.00$ | $87.50 \pm 2.21$ | $86.72 \pm 4.06$ |
| MMM2 | 50k | $18.68 \pm 3.42$ | $92.97 \pm 2.59$ | $62.50 \pm 11.69$ | $86.72 \pm 4.62$ |
| 2c_vs_64zg | 10k | $19.87 \pm 0.48$ | $99.22 \pm 1.35$ | $34.38 \pm 9.11$ | $92.97 \pm 2.59$ |
| corridor | 10k | $19.44 \pm 1.61$ | $96.88 \pm 3.83$ | $21.88 \pm 11.48$ | $94.53 \pm 2.59$ |
| 6h_vs_8z | 10k | $18.72 \pm 2.33$ | $75.00 \pm 5.85$ | $27.34 \pm 6.77$ | $76.56 \pm 6.44$ |
| bane_vs_bane | 10k | $19.61 \pm 1.26$ | $96.09 \pm 2.59$ | $91.41 \pm 4.62$ | $98.44 \pm 1.56$ |
| Average | $\sim$ | $19.20 \pm 2.04$ | $90.91 \pm 3.56$ | $51.78 \pm 7.27$ | $\mathbf{89.35} \pm 3.92$ |

the 11 training maps by utilizing the expert policies trained with a strong RL method named ACE (Li et al., 2022). This yields 11 offline datasets, most of which contain 10k episodes with an average return exceeding 18. Then, we employ different methods to pretrain on the offline dataset and evaluate their zero-shot capabilities on 60 generated test maps. As shown in Table 1, we run 32 test episodes to obtain the win rate and report the average win rate as well as the standard deviation across 4 seeds.

**Baseline** We take the MADT (Meng et al., 2023) as our baseline for comparison which utilizes a causal transformer to consider the history of local observation and action for an agent. MADT transforms multi-agent pretraining data into single-agent pretraining data and trains each agent's policy independently, therefore adopting a decentralized training and decentralized execution setting. Specifically, the input to each agent's policy is its own observation. Such an independent learning pipeline leads to an increase in computational complexity of $O\left(N^3\right)$ *w.r.t.* agent numbers $N$. Our experiments of MaskMA and baseline MADT utilize the same hyperparameters which are detailed in the Appendix.

## 4.1 Performance on Training Maps

Performance on the training domain is the most basic ability of one model, thus we first assess MaskMA and the baseline method on datasets including 11 training maps. As shown in Table 1, MaskMA achieves a 90.91% average win rate in 11 maps for Centralized Execution setting, while MADT can not be applied in this setting. Moreover, MaskMA surpasses MADT by a significant margin on Decentralized Execution setting (89.35% *v.s.* 51.78%). In some challenging maps like 3s5z_vs_3s6z, the advantage of our MaskMA is even much larger (85.16% *v.s.* 14.84%). One key observation is that MaskMA consistently performs well in both Centralized Training & Centralized Execution (CTCE) and Centralized Training & Decentralized Execution (CTDE) settings, highlighting its flexibility and adaptability in various execution paradigms. Figure 5a represents the curve of the average training win rate of MaskMA and the baseline method in 11 training maps on the Decentralized Execution setting. MaskMA significantly outperforms the baseline and achieves more than 80% win rate in most maps within 0.5M training steps, showing the robustness and efficiency of MaskMA.

## 4.2 MaskMA as Excellent Zero-shot Learners

Zero-shot generalization ability is the core motivation of our design, so we also evaluate the model's ability to generalize to extensive unseen scenarios *without* any retraining. Specifically, we evaluate our MaskMA and MADT on the 60 unseen testing maps. Table 2 shows that MaskMA outperforms MADT in zero-shot scenarios by a large margin in

Decentralized Execution setting (77.75% *v.s.* 43.72%), successfully transferring knowledge to new tasks without requiring any additional fine-tuning. In Centralized Execution setting, the performance of our MaskMA is even better with 79.71% win rate. These indicate that MaskMA's mask-based training strategy and generalizable action representation effectively address the challenges of partial observation, varying agent numbers, and different action spaces in multi-agent environments.

Furthermore, MaskMA consistently performs well across varying levels of environment complexity, as demonstrated by the win rates in different entity groups. In contrast, MADT's performance is obviously dropped in high-complexity environments (*i.e.*, # Entity > 20). This highlights the ability of MaskMA to generalize and adapt to diverse scenarios, which is a key feature of a robust multi-agent decision, making it a versatile and reliable choice for multi-agent tasks.

Table 2: **Zero-shot win rate on 60 unseen test maps** after training only on 11 training maps. "# Entity" denotes the number of entities present in each individual map, while "# Map" represents the number of maps fulfilling the entity condition. Our MaskMA outperforms MADT by a large margin in the *zero-shot* setting.

| # Entity | # Map | Centralized Execution | Decentralized Execution | |
| --- | --- | --- | --- | --- |
| | | MaskMA(Ours) | MADT | MaskMA(Ours) |
| $\leq 10$ | 23 | $76.26 \pm 3.30$ | $43.55 \pm 3.94$ | $74.38 \pm 3.57$ |
| $10 \sim 20$ | 22 | $83.81 \pm 2.85$ | $46.77 \pm 3.67$ | $80.08 \pm 2.98$ |
| $> 20$ | 15 | $79.01 \pm 5.02$ | $39.53 \pm 3.61$ | $79.48 \pm 3.84$ |
| All | 60 | $79.71 \pm 3.56$ | $43.72 \pm 3.76$ | $\textbf{77.75} \pm 3.42$ |

## 4.3 Performance on Downstream Tasks

Evaluating MaskMA's generalization ability to downstream tasks can help us understand whether the learned skills and strategies could be effectively adapted to different situations. In this section, we provide various downstream tasks to assess MaskMA's robust generalization, including varied policies of collaboration, ally malfunction, and ad hoc team play.

**Varied Policies Collaboration.** In this task, some agents are controlled by our trained policy (*i.e.*, controllable agents), while others are controlled by an external baseline policy (*i.e.*, uncontrollable agents). This task requires our trained policy to have excellent collaborative ability to cooperate with external policies. We conducted simulations in the 8m_vs_9m map, where our team controlled 8 marines to defeat 9 enemy marines. The uncontrollable agents are controlled by an external baseline policy with an average win rate of 41.41%. As shown in Table 3, MaskMA exhibits seamless collaboration with uncontrollable agents under different scenarios. MaskMA dynamically adapts to the strategies of other players and effectively coordinates actions. Specially, with only 50% controllable agents, the win rate has been improved significantly by nearly two times (79.69% *v.s.* 41.41%).

Table 3: **Win rate on varied policies collaboration task with 8m_vs_9m map.** The uncontrollable agents are controlled by a policy with 41.41% average win rate (*i.e.*, the win rate with 0% controllable agents). MaskMA demonstrates excellent collaborative performance in diverse scenarios with varying proportion of uncontrollable agents.

| Proportion of Our Controllable Agents | 0% | 25% | 50% | 75% | 100% |
| --- | --- | --- | --- | --- | --- |
| Win Rate | $41.41 \pm 6.00$ | $62.50 \pm 7.33$ | $79.69 \pm 5.18$ | $\textbf{89.84} \pm 2.59$ | $86.72 \pm 4.06$ |

**Ally Malfunction.** In this task, one ally marine may become disabled or die due to external factors during execution. The time of ally marine malfunction is a changeable parameter. MaskMA is requested to handle such situations gracefully by redistributing tasks among the remaining agents while maintaining overall performance. Simulations are conducted in the 8m_vs_9m map, where we control 8 marines against 9 enemy marines. The average win rate without ally malfunction is 86.72%. As given in Table 4, MaskMA exhibits robustness and adaptability in the face of unexpected ally malfunction.

Table 4: **Win rate on ally malfunction task with 8m_vs_9m map.** "Marine Malfunction Time" indicates the time of a marine malfunction during an episode. For instance, 0.2 means that one ally marine begins to exhibit a stationary behavior at $1/5^{th}$ of the episode. "None" signifies the original 8m_vs_9m configuration without any marine malfunctions.

| Ally Marine Malfunction Time | None | 0.2 | 0.4 | 0.6 | 0.8 |
|---|---|---|---|---|---|
| Win Rate | $86.72 \pm 4.06$ | $1.56 \pm 1.56$ | $37.5 \pm 6.99$ | $71.09 \pm 6.77$ | $86.72 \pm 2.59$ |

**Ad Hoc Team Play.**   In this task, agents need to quickly form a team with one new ally agent during the execution of the task. The challenge lies in the ability of the model to incorporate this new agent into the team and allow them to contribute effectively without disrupting the ongoing strategy. Simulations are conducted in the 7m_vs_9m map (*i.e.*, our 7 marines to defeat 9 enemy marines), and the average win rate without extra ally marine is 0%. As shown in Table 5, MaskMA demonstrates excellent performance in ad hoc team play scenarios, adjusting its strategy to accommodate new agents and ensuring a cohesive team performance. We provide visualizations of MaskMA's behavior on ad hoc team play in Figure 4. More results are presented in the Appendix.

Table 5: **Win rate on ad hoc team play task with 7m_vs_9m map.** "Marine Inclusion Time" indicates the time of adding an extra ally marine during an episode. For example, 0.2 represents adding one ally marine at $1/5^{th}$ of the episode. "None" signifies the original 7m_vs_9m setup without any extra ally marine.

| Ally Marine Inclusion Time | None | 0.2 | 0.4 | 0.6 | 0.8 |
|---|---|---|---|---|---|
| Win Rate | $0 \pm 0$ | $80.47 \pm 7.12$ | $78.12 \pm 2.21$ | $50.00 \pm 8.84$ | $10.94 \pm 6.81$ |

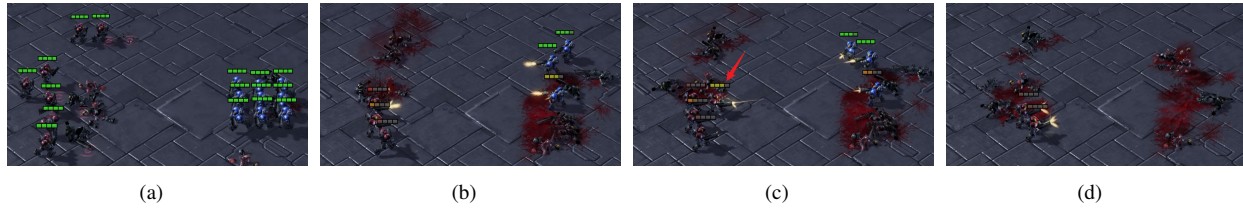

|     (a)     |     (b)     |     (c)     |     (d)     |

Figure 4: **Visualization of Ad Hoc Team Play** on 7m_vs_9m with Marine Inclusion Time = 0.8. This experiment demonstrates that when new Marines are added near the end of an episode, MaskMA still can quickly incorporate them into the team and enable them to contribute effectively. **(a)** Initial distribution of agents' positions. **(b)** Prior to the addition of the new Marine, our team is left with only three severely wounded agents, on the brink of defeat. **(c)** The new agent (indicated by the red arrow) joins our team and immediately engages the enemy. **(d)** With the assistance of the newly added agent, our team successfully defeats the enemy.

## 4.4   Ablation Study

We perform extensive ablation studies to verify the effectiveness of proposed components and hyper-parameter choices.

**Generalizable Action Representation (GAR).**   We compare GAR module to an alternative action representation method, *i.e.*, aligning the maximum action length with a specific action mask for each task. As shown in Table 6, removing GAR leads to significant performance degradation (#2 *v.s.* #4) on the testing maps, emphasizing its importance in improving the model's generalization capabilities.

**Mask-based Training Strategy (MTS).**   The ablation results are shown in Table 6. The alternative method of MTS is centralized training without masking, which naturally excels in the Centralized Execution setting (#3). However, the model without MTS struggles to generalize to Decentralized Execution setting, exhibiting significant performance degradation (#3 *v.s.* #4). The MTS employed in MaskMA, while posing a challenging training task, helps the model learn ~~permanent representations~~ stable representations across different scenarios and observation range. Intuitively, the random mask ratio aligns with the inference process, where the number of enemies and allies gradually increases in an agent's local observation due to cooperative micro-operations, *e.g.,* positioning, kiting, and focusing fire.

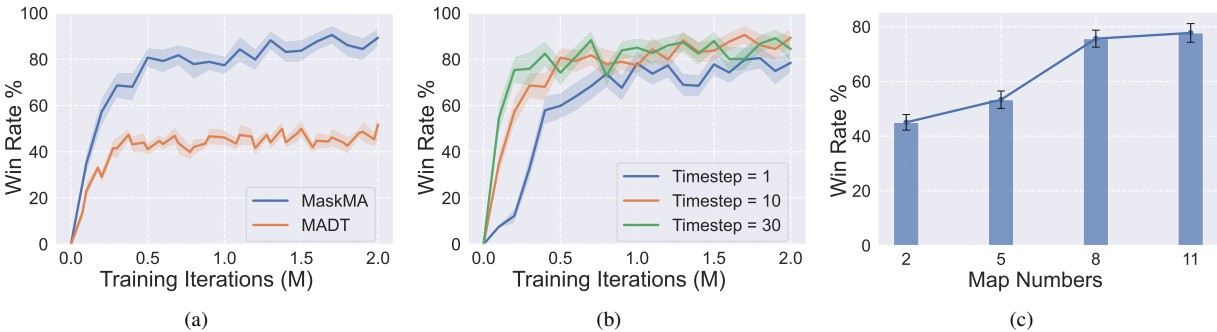

Figure 5: **(a) Comparison of learning curve** with the win rate on training maps in Decentralized Execution setting. MaskMA consistently outperforms MADT on average win rate. **(b) Ablation on timestep** with the win rate on training maps in the Decentralized Execution setting. MaskMA performs better with a longer timestep. **(c) Ablation on the number of training maps** with the win rate on unseen maps in the Decentralized Execution setting. With the increasing of training maps (especially from 5 to 8), the model's performance on various unseen maps improves, indicating better generalization ability.

Table 6: **Ablation over mask-based training strategy (MTS) and generalizable action representation (GAR).** The results are the performance on training maps. The baseline is a naive transformer. Each row adds a new component to the baseline, showcasing how each modification affects the performance.

| # | Module | | Setting | |
|---|---|---|---|---|
| | MTS | GAR | Cent. Exe. | Decent. Exe. |
| 1 | | | $44.67 \pm 3.35$ | $8.03 \pm 1.44$ |
| 2 | ✓ | | $39.49 \pm 3.05$ | $39.91 \pm 3.97$ |
| 3 | | ✓ | $91.26 \pm 4.21$ | $41.55 \pm 4.38$ |
| 4 | ✓ | ✓ | $90.91 \pm 3.56$ | $89.35 \pm 3.92$ |

**Mask Ratio in MTS.** We explore different types of masks for training, including a set of fixed mask ratios, local mask, and random mask ratios chosen from $[0, 1]$ for units at each time step. We provide mask ratio analysis in Table 7. The results show that as the masking ratio increases, the performance on the training maps increases in the Decentralized Execution setting while decreasing in the Centralized Execution setting. This suggests that an appropriate masking ratio helps strike a balance between learning useful representations and maintaining adaptability to dynamic scenarios in the agent's local observation. In conclusion, a random mask ratio with the range of $[0, 1]$ is a simple yet effective way to balance the two settings, which could combine the advantages of various ratio masks. This approach allows MaskMA to demonstrate strong performance in both centralized and decentralized settings while maintaining the adaptability and generalization necessary for complex multi-agent tasks.

**Timestep Length.** We conduct experiments with different timestep length to study the access to previous states. As shown in Figure 5b, MaskMA's performance is better when using a longer timestep length. One hypothesis is that the POMDP property of the SMAC environment necessitates that policies in SMAC take into account sufficient historical information to make informed decisions. Considering the balance between performance and efficiency, we select timestep $K$ as 10 as our default hyper-parameter. This choice allows MaskMA to leverage enough historical information to make well-informed decisions while maintaining a reasonable level of computational complexity.

**Number of Training Maps.** Figure 5c demonstrates the influences of the number of training maps. As the number of training maps increases, the win rate on the testing maps also improves, indicating that the model is better equipped to tackle new situations. A marked uptick in win rate is observed when the map count rises from 5 to 8, underlining the value of training the model across varied settings. This trend in MaskMA offers exciting prospects for multi-agent decision-making. It implies that by augmenting the count of training maps or integrating richer, more intricate training scenarios, the model can bolster its adaptability and generalization skills.

Table 7: **Ablation study of mask ratio for training.** "Local Mask" represents using the local visibility of the agent as the mask.

| Setting | Local Mask | Mask Ratio | | | | |
|---|---|---|---|---|---|---|
| | | 0 | 0.2 | 0.5 | 0.8 | Random [0, 1] |
| Centralized Execution | $55.97 \pm 4.67$ | $\mathbf{91.26} \pm 4.21$ | $89.70 \pm 3.81$ | $88.21 \pm 3.78$ | $82.81 \pm 4.83$ | $90.91 \pm 3.56$ |
| Decentralized Execution | $83.59 \pm 8.08$ | $41.55 \pm 4.38$ | $58.03 \pm 5.70$ | $71.52 \pm 4.23$ | $82.03 \pm 5.01$ | $\mathbf{89.35} \pm 3.92$ |

**Training Cost and Parameter Count.**    MaskMA processes the inputs of all agents concurrently, achieving a notable degree of parallelism superior to MADT, which transforms multi-agent training data into single-agent data. Consequently, MaskMA is considerably more time-efficient than MADT when trained over identical epochs. Specifically, training 2M iterations on one A100 GPU with the same model architecture and the same 190K training data, MaskMA needs only 31 hours, whereas MADT requires 70 hours. Also, the parameter counts for both models are nearly identical since they share the same transformer architecture and the only difference lies in the final fully connected (FC) layer for action output.

## 5    Discussion

We have introduced MaskMA, a mask-based collaborative learning framework for multi-agent decision-making, to tackle the challenges of zero-shot generalization in the multi-agent decision-making field. MaskMA consists of a transformer architecture, a mask-based training strategy, and a generalizable action representation. Extensive experiments on SMAC dataset demonstrate the effectiveness of our MaskMA in terms of zero-shot performance and adaptability to various downstream tasks, such as varied policies collaboration, ally malfunction, and ad hoc team play.

**Limitations & Future Work**   Our current study primarily relies on expert datasets. Exploring how to utilize low-quality demonstration datasets to develop a generalist agent with robust zero-shot capabilities presents a promising research direction. Furthermore, extending the maskMA approach to create a single generalist agent capable of generalizing across different types of environments, such as entirely different games, is also a promising avenue for future research.

## Acknowledgment

This work was supported by the JC STEM Lab of AI for Science and Engineering, funded by The Hong Kong Jockey Club Charities Trust.

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

In this Supplementary Material, we provide more elaboration on the implementation details and experiment results. Specifically, we present the implementation details of the model training in Section A and additional results and visualization in Section B.

# A    Additional Implementation Details

In this section, we provide a detailed description of the required environment, hyperparameters.

## A.1    Environment.

We use the following software versions:

- CentOS 7.9, Python 3.8.5, Pytorch 2.0.0 (Paszke et al., 2019)

- StarCraft II 4.10 (Samvelyan et al., 2019), DI-engine 0.2.0 (engine Contributors, 2021)

We conduct all experiments with a single A100 GPU.

## A.2    Hyperparameters

As shown in Table 8, our experiments of MaskMA and baseline MADT (Meng et al., 2023) utilize the same hyperparameters.

Table 8: Hyperparameters of MaskMA and MADT. It should be noted that both models utilize the exact same set of hyperparameters.

|  | Hyperparameter | Value |
|---|---|---|
| Training | Optimizer | RMSProp |
|  | Learning rate | 1e-4 |
|  | Batch size | 256 |
|  | Weight decay | 1e-5 |
| Architecture | Number of blocks | 6 |
|  | Hidden dim | 128 |
|  | Number of heads | 8 |
|  | Timestep length | 10 |

## A.3    Differences between the training and testing maps

We list some differences between the training and testing maps in Table 9.

| Difference | Testing Maps | Most Similar Training Maps |
|---|---|---|
| Number of agents | 10m_vs_11m | 5m_vs_6m |
| Different race matchups | Protoss vs. Terran | Protoss vs. Protoss, Terran vs. Terran |
| Reversed ally and enemy | 64zg_vs_2c | 2c_vs_64zg |

Table 9: Comparison of testing maps and training maps

## A.4    Pseudocode

The Pytorch-style implementation of the MaskMA is shown below:

```
# c: controllable, uc: uncontrollable
# Input: state_embeddings for c_entity [s_1, ..., s_M] and uc_entity [s_M+1, ..., s_N]
# Initial: func: GenerateMask

# Enhance state_embeddings with information of timestep and controllable
state_embedding_c = state_embedding_c + timestep_embedding + controllable_embedding
state_embedding_uc = state_embedding_uc + timestep_embedding + uncontrollable_embedding

# Mask-based Training Strategy
# Generate random mask
mask = GenerateMask(mask_ratio, state_embedding_c, state_embedding_uc)
# Apply masks and process through the transformer
hidden_state_c, hidden_state_uc = transformer(state_embedding_c,
                                               state_embedding_uc, mask)

actions = GAR(hidden_state_c, hidden_state_uc)

# Input: hidden_state_c [B, T, M, D], hidden_state_uc [B, T, N-M, D]
# Initial: FC layers: fc_interactive, fc_intrinsic
def GAR(hidden_state_c, hidden_state_uc):
    # Interactive actions
    hidden_state = torch.cat([hidden_state_c, hidden_state_uc], dim=-2)
    # Concatenate and repeat embeddings of hidden state for prediction
    hidden_state_c_repeat = hidden_state_c.unsqueeze(3).repeat(1, 1, 1, N, 1)
    hidden_state_repeat = hidden_state.unsqueeze(2).repeat(1, 1, M, 1, 1)
    interactive_action_embedding = torch.cat([hidden_state_c_repeat,
                                              hidden_state_repeat], dim=-1)

    # Process result through attack_action network
    interactive_logits = fc_interactive(interactive_action_embedding)

    # Intrinsic actions
    intrinsic_action_embedding = hidden_state_c
    # Compute intrinsic logits using action queries
    intrinsic_logits = fc_intrinsic(intrinsic_action_embedding)

    # Final actions
    # Concatenate logits to form action predictions
    action_preds =  torch.cat([interactive_logits, intrinsic_logits], dim=-1)
    # Determine final actions by selecting the maximum valid logits
    actions = action_preds.max(-1).indices
    return actions
```

## A.5 State of units

In the original StarCraft II Multi-Agent Challenge (SMAC) setting, the length of the observation feature fluctuates in accordance with the number of agents. To enhance generalization, MaskMA directly utilizes each unit's state as input to the transformer architecture. As depicted in Table 10, within the SMAC context, each unit's state comprises 42 elements, constructed from nine distinct sections. Specifically, the unit type section, with a length of 10, represents the nine unit types along with an additional reserved type.

# B Additional Results and Analysis

## B.1 Win rate of all maps

As shown in Table 11, we present the win rate of MaskMA and MADT on 11 training maps and 60 testing maps.

Table 10: The composition of the state.

| State name | dim |
|---|---|
| ally or enemy | 1 |
| unit type | 10 |
| pos.x and y | 2 |
| health | 1 |
| shield | 2 |
| cooldown | 1 |
| last action | 7 |
| path | 9 |
| height | 9 |

## B.2  Comparition of pretraining dataset size

The scale of the pretraining dataset significantly impacts the eventual performance. In the multi-agent StarCraft II environment, SMAC, we investigated the optimal size for the pretraining dataset. We take the 3s_vs_5z map as an example and solely use the pretraining dataset of this map to train MaskMA, and then test it on the same map. As illustrated in Figure 6, a dataset encompasing 1k episodes was found insufficient, leading to a progressive decline in win rates. In contrast, a dataset comprising 50k episodes demonstrated exceptional performance. Specifically, for the 3s_vs_5z and MMM2 maps, a pretraining dataset containing 50k episodes proved appropriate. For the remaining nine maps, a dataset consisting of 10k episodes was found to be suitable.

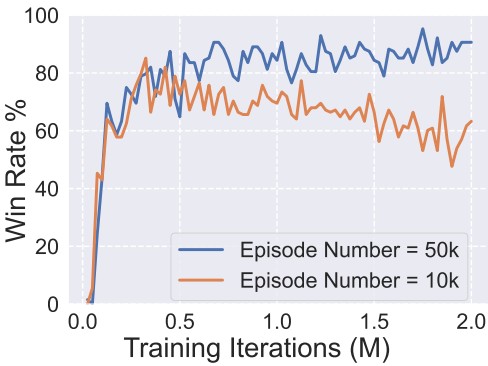

Figure 6: Performance comparison across different sizes of the pretraining dataset on the 3s_vs_5z Map.

## B.3  Visualization

In this section, we provide visualizations of MaskMA's behaviors on additional two downstream tasks: varied policies collaboration, and teammate malfunction. Figure 7, 8 evaluate the strong generalization of MaskMA. Additionally, we offer replay videos for a more comprehensive understanding of MaskMA's behavior and strategies.

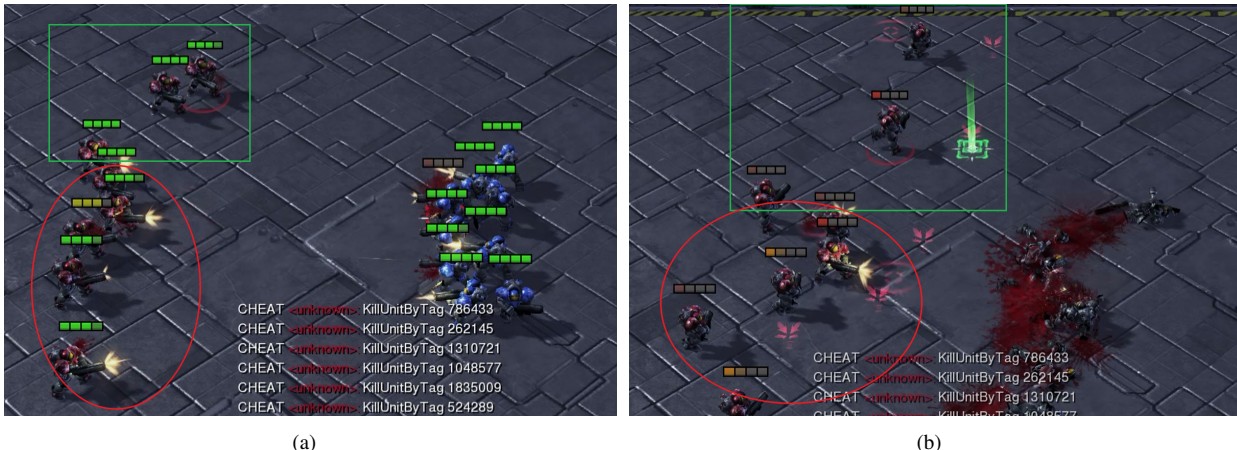

Figure 7: Varied Policies Collaboration on 8m_vs_9m with 4 Agents with different policies. The agents within the green box are controlled by other policies (replaced with a network trained with a 41% win rate), while the agents within the red circle are controlled by MaskMA. **(a)**: Initial distribution of agents' positions. **(b)**: MaskMA dynamically adapts to the strategies of players by different policies and effectively coordinates actions, resulting in a victorious outcome.

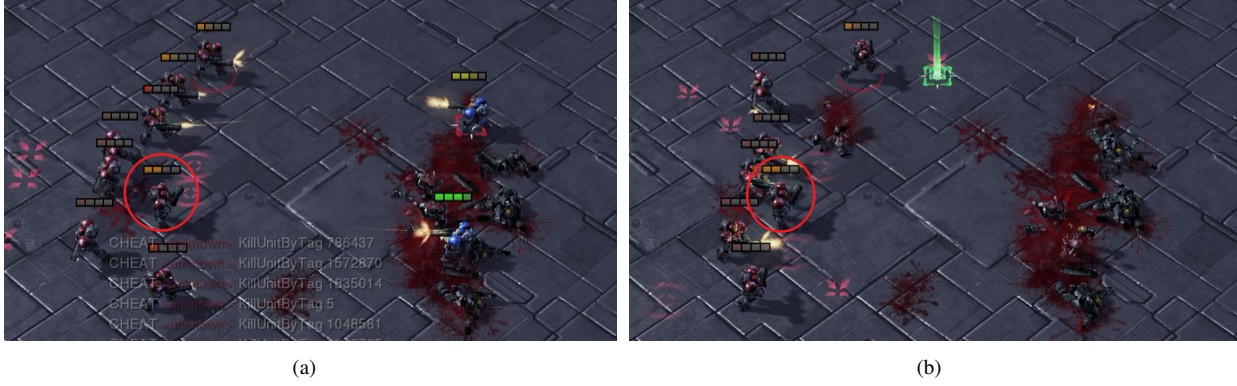

Figure 8: Teammate Malfunction on 8m_vs_9m with Marine Malfunction Time = 0.6. The agent within the red circle suddenly malfunctions in the middle of the episode, remaining stationary and taking no actions. **(a)**: The agent within the red circle starts malfunctioning. **(b)**: Despite the malfunctioning teammate, other agents continue to collaborate effectively and eventually succeed in eliminating the enemy.

Table 11: Win rate of MaskMA and MADT on 11 training maps and 60 testing maps.

| Map | Ours | | MADT |
|---|---|---|---|
| | Centralized Execution | Decentralized Execution | Decentralized Execution |
| 3s_vs_5z | 85.94 ±3.49 | 82.81 ±7.81 | 73.44 ±3.49 |
| 3s5z | 98.44 ±1.56 | 99.22 ±1.35 | 15.62 ±6.99 |
| 1c3s5z | 94.53 ±4.06 | 95.31 ±1.56 | 54.69 ±8.41 |
| 3s5z_vs_3s6z | 85.94 ±6.44 | 85.16 ±5.58 | 14.84 ±9.97 |
| 5m_vs_6m | 86.72 ±1.35 | 84.38 ±4.94 | 85.94 ±5.18 |
| 8m_vs_9m | 88.28 ±6.00 | 86.72 ±4.06 | 87.50 ±2.21 |
| MMM2 | 92.97 ±2.59 | 86.72 ±4.62 | 62.50 ±11.69 |
| 2c_vs_64zg | 99.22 ±1.35 | 92.97 ±2.59 | 34.38 ±9.11 |
| corridor | 96.88 ±3.83 | 94.53 ±2.59 | 21.88 ±11.48 |
| 6h_vs_8z | 75.00 ±5.85 | 76.56 ±6.44 | 27.34 ±6.77 |
| bane_vs_bane | 96.09 ±2.59 | 98.44 ±1.56 | 91.41 ±4.62 |
| 2s_vs_1z | 100.00±0.00 | 100.00±0.00 | 70.31 ±10.00 |
| 3z_vs_3s | 71.88 ±3.83 | 53.91 ±9.47 | 0.00 ±0.00 |
| 1c2s | 44.53 ±5.58 | 41.41 ±6.39 | 2.34 ±2.59 |
| 4z_vs_3s | 99.22 ±1.35 | 97.66 ±1.35 | 4.69 ±2.71 |
| 4m | 99.22 ±1.35 | 100.00±0.00 | 28.91 ±6.00 |
| 1c3s | 54.69 ±3.49 | 48.44 ±4.69 | 17.19 ±4.69 |
| 5m_vs_4m | 100.00±0.00 | 100.00±0.00 | 100.00±0.00 |
| 5m | 100.00±0.00 | 100.00±0.00 | 98.44 ±1.56 |
| 2s3z | 10.94 ±5.18 | 8.59 ±2.59 | 0.00 ±0.00 |
| 1s4z_vs_1ma4m | 91.41 ±4.62 | 88.59 ±3.41 | 35.16 ±3.41 |
| 3s2z_vs_2s3z | 9.38 ±5.85 | 11.72 ±2.59 | 0.00 ±0.00 |
| 2s3z_vs_1ma2m2me | 92.97 ±4.06 | 82.03 ±14.04 | 35.16 ±9.73 |
| 2s3z_vs_1ma3m1me | 90.62 ±3.12 | 93.75 ±3.83 | 37.50 ±7.97 |
| 1c3s1z_vs_1ma4m | 100.00±0.00 | 99.22 ±1.35 | 98.44 ±1.56 |
| 1s4z_vs_2ma3m | 83.59 ±7.12 | 77.66 ±3.41 | 73.44 ±6.44 |
| 1c3s1z_vs_2s3z | 100.00±0.00 | 100.00±0.00 | 78.12 ±5.85 |
| 2s3z_vs_1h4zg | 96.88 ±3.12 | 96.09 ±3.41 | 71.88 ±7.16 |
| 3s2z_vs_2h3zg | 95.31 ±5.18 | 93.75 ±4.42 | 75.00 ±3.83 |
| 1c4z_vs_2s3z | 100.00±0.00 | 100.00±0.00 | 71.88 ±6.63 |
| 2ma3m_vs_2ma2m1me | 46.09 ±12.57 | 39.06 ±9.24 | 0.00 ±0.00 |
| 2s3z_vs_1b1h3zg | 100.00±0.00 | 96.88 ±2.21 | 67.19 ±4.06 |
| 1s4z_vs_5z | 3.12 ±2.21 | 5.47 ±4.06 | 0.00 ±0.00 |
| 1c1s3z_vs_1c4z | 64.06 ±7.16 | 76.56 ±5.63 | 35.94 ±6.44 |
| 6m | 97.66 ±2.59 | 99.22 ±1.35 | 47.66 ±5.12 |
| 1s3z_vs_5b5zg | 93.75 ±3.83 | 87.50 ±2.21 | 58.59 ±1.35 |
| 8z_vs_6h | 93.75 ±2.21 | 98.44 ±1.56 | 38.28 ±4.06 |
| 2s5z | 17.97 ±4.06 | 7.03 ±3.41 | 0.78 ±1.35 |
| 1c3s1z_vs_1b1h8zg | 100.00±0.00 | 100.00±0.00 | 57.81 ±0.00 |
| 1c3s1z_vs_1h9zg | 100.00±0.00 | 100.00±0.00 | 60.94 ±0.00 |
| 2c1s2z_vs_1h9zg | 100.00±0.00 | 100.00±0.00 | 49.22 ±0.00 |
| 3ma6m1me_vs_3h2zg | 100.00±0.00 | 100.00±0.00 | 23.44 ±0.00 |
| 3s2z_vs_1b1h8zg | 97.66 ±1.35 | 85.94 ±4.69 | 52.34 ±2.59 |
| 5ma4m1me_vs_1b3h1zg | 100.00±0.00 | 100.00±0.00 | 98.44 ±2.71 |
| 4ma5m1me_vs_5zg | 100.00±0.00 | 100.00±0.00 | 85.94 ±8.12 |
| 1s5z_vs_4ma6m | 39.06 ±10.00 | 16.41 ±3.41 | 4.69 ±4.69 |
| 4ma7m_vs_2s3z | 98.44 ±1.56 | 100.00±0.00 | 35.16 ±13.07 |
| 2ma9m_vs_3s2z | 89.06 ±2.71 | 80.47 ±6.00 | 51.56 ±9.50 |
| 5s2z_vs_2ma8m | 48.44 ±9.24 | 29.69 ±11.16 | 8.59 ±4.06 |
| 2s5z_vs_9m1me | 99.22 ±1.35 | 99.22 ±1.35 | 46.09 ±4.94 |
| 1c3s_vs_3h10zg | 85.94 ±5.63 | 89.84 ±5.58 | 43.75 ±4.94 |
| 3s7z_vs_5ma2m3me | 99.22 ±1.35 | 96.88 ±3.83 | 60.94 ±2.59 |
| 1c3s6z | 58.59 ±7.12 | 61.72 ±7.77 | 14.84 ±6.77 |
| 2c2s6z_vs_1c3s6z | 100.00±0.00 | 100.00±0.00 | 97.66 ±1.35 |
| 1c2s7z | 25.00 ±9.63 | 25.78 ±5.12 | 7.81 ±3.49 |
| 1c4s5z_vs_5ma3m2me | 100.00±0.00 | 83.59 ±8.08 | 84.38 ±0.00 |
| 10s1z_vs_1b2h7zg | 96.09 ±4.06 | 96.88 ±2.21 | 68.75 ±1.56 |
| 10m_vs_11m | 79.69 ±6.44 | 78.91 ±7.45 | 0.00 ±0.00 |
| 1b10zg | 85.16 ±5.58 | 89.06 ±7.16 | 60.16 ±5.12 |
| 2b10zg | 92.19 ±4.69 | 98.44 ±1.56 | 30.47 ±5.58 |
| 1c_vs_32zg | 59.38 ±9.38 | 55.47 ±7.77 | 1.56 ±2.71 |
| 32zg_vs_1c | 92.97 ±3.41 | 94.53 ±3.41 | 53.91 ±3.12 |
| 1b20zg | 74.22 ±10.91 | 82.81 ±2.71 | 66.41 ±7.45 |
| 2b20zg | 100.00±0.00 | 97.66 ±1.35 | 71.09 ±4.94 |
| 3b20zg | 96.09 ±3.41 | 88.28 ±4.62 | 85.94 ±4.69 |
| 16z_vs_6h24zg | 97.66 ±2.59 | 97.66 ±2.59 | 71.09 ±4.06 |
| 5b20zg | 99.22 ±1.35 | 100.00±0.00 | 32.03 ±2.59 |
| 64zg_vs_2c | 55.47 ±6.39 | 56.25 ±2.21 | 48.44 ±8.41 |
| 1c8z_vs_64zg | 89.06 ±7.16 | 85.94 ±5.18 | 2.34 ±2.59 |
| 1c8z_vs_2b64zg | 61.72 ±7.77 | 65.62 ±5.85 | 0.00 ±0.00 |
| 1c8z_vs_5b64zg | 6.25 ±2.21 | 4.69 ±3.49 | 0.78 ±1.35 |

