# OpenReview forum: "MaskMA: Towards Zero-Shot Multi-Agent Decision Making with Mask-Based Collaborative Learning"
_TMLR — Accepted by TMLR_

### Review · Reviewer_YL12 · 2024-04-12

**Summary Of Contributions:**

This paper proposes a new method for multi-agent sequential decision problems based on imitation learning using the transformer architecture. In particular, the authors use the Multi-Agent Decision Transformer framework proposed in prior work and modify it to better capture the specifics of multi-agent decision problems. In particular, they introduce mask-based training, which allows for centralized training and decentralized execution, and generalizable action representations, which allow the model to better handle interactive actions. The authors test the proposed approach on the StarCraft Multi-Agent Challenge with various scenarios and variations, showing that MaskMA outperforms MADT. They also run several ablation studies to identify where the improvements come from.

**Audience:**

Yes

**Claims And Evidence:**

No

**Requested Changes:**

- Please address the issues with the presentation mentioned above.
- Could you provide a hypothesis for why GAR works so well? In particular, I'm curious to understand what it can do that MADT (or a transformer in general) can't.

**Strengths And Weaknesses:**

Strengths:
- The problem considered in the paper is important. Connecting the sequence modeling literature with multi-agent decision-making is an interesting avenue of research.
- The empirical results presented in the paper are quite strong. The proposed improvements do seem to matter a lot.
- The paper offers valuable ablation studies and analysis of interesting tasks, such as combining neural agents with scripted agents, introducing new agents during the episode, etc.

Weaknesses:
- There are some issues with the presentation:
    - GAR is poorly described. This is a central part of the paper but the idea is only explained in two paragraphs and a part of a figure. After reading the paper carefully I don't have a precise understanding of how it works. Why are there as many actions as there are states in the current timestep? I would appreciate formal equations or a pseudocode that explains GAR.
    - The notion of interactive and intrinsic actions is not very well described. Can we provide a general formulation in the MDP framework about what “interactive” means? Having an explicit list of interactive/intrinsic actions in SMAC would be useful. I looked briefly at other papers and I haven't found definitions there.
    - Some of the citations look broken, e.g., the [Carroll et al.] one does not include the year.
- Also, I don't think the paper explains why GAR works so well. In particular, the transformer is already able to combine the embeddings across different agents through attention. Would it be possible to implement GAR using a single (or a few) attention layers? If not, then what part is specifically impossible for attention? If yes, why GAR helps?
- Having more baselines that are standard in SMAC would be nice. For example, the MADT paper also includes BC, BCQ-MA, CQL-MA, and ICQ-MA. Evaluating some of the papers mentioned in the related work would also be beneficial. This is more of a "nice-to-have" than a major weakness. I would be also okay with mentioning this point in the limitations section.

---

> ### Author Response · Authors · 2024-07-09
> **Response to Reviewer YL12**
>
> Thank you for your valuable comments and thorough suggestions. We have addressed all identified weaknesses and revised the manuscript, highlighting the modifications in blue according to the required changes. Please review the updated manuscript for detailed changes.
>
> ### Weaknesses
>
> > W1. GAR is poorly described. Why are there as many actions as there are states in the current timestep?
> >
>
> Thank you very much for your suggestion. We have added a PyTorch-style **pseudocode** of our method in Appendix A.4 in the revision. For the 'state and action number' questions, we provide a multi-agent formulation preliminaries in Section 3.1.1. Specifically, there are multiple controllable agents for multi-agent tasks. The global state at the $t$-th time step is defined as $\textbf{s}^{t}=(s^t_1, s^t_2, ..., s^t_N)$, where each $s^t_i$ exclusively represents the state of agent $u_i$ at timestep $t$. Each agent $u_i$ selects an action $a^t_i \in A_i$, forming a joint action $\boldsymbol{a}^t \in \mathbf{A}$.
>
> ------
>
> > W2. The notion of interactive and intrinsic actions is not very well described.
> >
>
> We are sorry for not clearly defining "interactive" and "intrinsic" actions. To provide clarity:
>
> - Intrinsic Actions: These are actions taken by an agent that do not depend on other agents. They are typically decisions that an agent makes based solely on its local observation or state. For example, in SMAC, intrinsic actions include moving up, down, left, or right.
> - Interactive Actions: In contrast, interactive actions depend on other agents. Examples of interactive actions in SMAC include attacking an enemy or healing an ally.
>
> ------
>
> > W3. Some of the citations look broken, e.g., the [Carroll et al.] one does not include the year.
> >
>
> We are so sorry for these typos. We have fixed them in the revised version.
>
> ------
>
> > W4. Why GAR works so well. In particular, the transformer is already able to combine the embeddings across different agents through attention. I'm curious to understand what it can do that MADT (or a transformer in general) can't.
> >
>
> We apologize for the misunderstanding. GAR is not designed to combine the embeddings across different agents like attention layers.
>
> GAR is designed to effectively manages varying action spaces from different maps by decoupling them into interactive and intrinsic actions, whereas attention layers often struggle with this due to their inherent design limitations.
>
> Specifically, MADT (or transformers in general) processes varying action spaces by padding them to the maximum action space observed across all scenarios, which can miss the specific context of each action. For instance, in mixed training scenarios like 5m_vs_6m (action space 4+6) and 3s_vs_5z (action space 4+5), the initial part of the action sequence denotes intrinsic actions such as moving up, down, left, or right, while the latter part denotes attacking a specific agent. For 5m_vs_6m, action 5 means attack the first enemy Marine, while action 5 in 3s_vs_5z means attack the first enemy Zealot. **The same action represents different physical meanings**. Additionally, this design **cannot** generalize to larger action spaces, such as 6m_vs_7m (4+7), where action 11 (attack the 7th enemy) is meaningless and out of the training scope. Therefore, this design of action spaces is not suitable for multi-agent pretraining.
>
> ------
>
> > W5. Having more baselines.
> >
>
> Thank you for your suggestions. We only adopted MADT as the baseline method because MADT is more powerful than BC, BCQ-MA, CQL-MA, and ICQ-MA, as stated in MADT paper[1].
>
> [1] Meng, Linghui, et al. "Offline pre-trained multi-agent decision transformer." *Machine Intelligence Research* 20.2 (2023): 233-248.
>
> ------
>
> ### Requested Changes
>
> > Please address the issues with the presentation mentioned above.
> >
>
> Thank you very much for your meticulous review of our manuscript. We have addressed all the requested changes in the revised version of the manuscript.
>
> ------
>
> > Could you provide a hypothesis for why GAR works so well? In particular, I'm curious to understand what it can do that MADT (or a transformer in general) can't.
> >
>
> Please see the response to W4.
>
> ------
>
> Thank you again for your valuable suggestions and for helping us enhance our work.

---

### Review · Reviewer_HV2H · 2024-06-05

**Summary Of Contributions:**

A Mask-Based collaborative learning framework for Multi-Agent decision making is proposed. A self-supervised learning scheme is discussed with the purpose to establish the generalization ability. The authors verify the ability of this aspect Through some StarCraft II experiments.

**Audience:**

Yes

**Broader Impact Concerns:**

no concerns

**Claims And Evidence:**

Yes

**Requested Changes:**

More complicated scene experiments need to be discussed, especially the authors should prove or measure the "zero-shot" distance between scenarios.
Based on these results, the effectiveness of the method proposed by the method can be verified much more convincing.

**Strengths And Weaknesses:**

Strengths:
1. Good research problem and motivation, i.e., the zero-shot ability of MARL;
2. The framework of the algorithm is clear;

Weaknesses:
The experiments proposed in the paper are constructive. However although the experimental scenarios of StarCraft II look rich, the similarity between StarCraft II scenes is very large which weakens the final conclusion.

---

> ### Author Response · Authors · 2024-07-09
> **Response to Reviewer HV2H**
>
> Thank you for your valuable comments. We have addressed weaknesses and revised the manuscript, highlighting the modifications in blue according to the required changes. Please review the updated manuscript for detailed changes.
>
> ### Weakness
>
> > W1. The experiments proposed in the paper are constructive. However although the experimental scenarios of StarCraft II look rich, the similarity between StarCraft II scenes is very large which weakens the final conclusion.
> >
>
> Thank you for this great suggestion. We focus on the SMAC domain because it is one of the most widely used environments in the cooperative multi-agent field. Although there is a similarity between StarCraft II scenes, building a generalist model capable of zero-shot transfer to new maps, including those with different maps and varying numbers of agents, remains an open challenge. The previous state-of-the-art method, MADT, only achieves a 44% zero-shot win rate in our SMAC experimental scenarios.
>
> For multi-agent pretraining, a critical problem is policy transfer. For example, if we train a policy using the 3s_vs_5z and 2ma7m1me_vs_3ma8m1me maps, we test it on the 1s4z_vs_2ma3m and 2s3z_vs_1ma3m1me maps to assess policy transfer. Agents must adeptly execute skills such as alternating fire, kiting, focus fire, and positioning to secure victory, making zero-shot transfer profoundly challenging.
>
> We understand the importance of testing our method in diverse environments to comprehensively assess its adaptability. However, we believe that validation across unseen and different maps within SMAC remains significant.
>
> ------
>
>
> ### Requested Changes
>
> > More complicated scene experiments need to be discussed, especially the authors should prove or measure the "zero-shot" distance between scenarios. Based on these results, the effectiveness of the method proposed by the method can be verified much more convincing.
> >
>
> Thank you for your suggestions. We have listed some differences between the training and testing maps in the table below, and have included this table in Appendix A.3 of the revised version.
>
> | Difference | Testing Maps | Most Similar Training Maps |
> | --- | --- | --- |
> | Number of agents | 10m_vs_11m | 5m_vs_6m |
> | Different race matchups | Protoss vs. Terran | Protoss vs. Protoss , Terran vs. Terran |
> | Reversed ally and enemy | 64zg_vs_2c | 2c_vs_64zg |
> | … |  |  |
>
> In fact, the "zero-shot" distance between scenarios in our experiments is already significant. The previous state-of-the-art method, MADT, only achieves a 44% zero-shot win rate in the testing maps.
>
> Thank you again for your valuable suggestions and for helping us enhance our work.

---

> > ### Comment · Reviewer_HV2H · 2024-07-15
> > **To authors**
> >
> > Thank you for your answers. I have carefully read your responses to my questions. The supplementary experiment discussions have proved the effectiveness of your method to a certain extent and can be considered as the results of the strengthened paper.

---

### Review · Reviewer_rbcy · 2024-06-24

**Summary Of Contributions:**

This paper makes two main contributions for cooperative multi-agent decision-making in the context of transformer-based policies: 1) random masking to improve generalization when the number of agents changes from train to test; 2) handling variable number of agents by using a fully-connected layer with output dimension 1 to compute the logits of interaction actions. Empirical results and ablations in the SMAC benchmark show that these two methods lead to improvement in zero-shot generalization over a previous transformer-based method, are robust to environment perturbations (variable number of policy-controlled agents, ally malfunction, ad-hoc team play).

**Audience:**

Yes

**Claims And Evidence:**

Yes

**Requested Changes:**

* [critical] Revise statements that imply the novelty of the Generalizable Action Representation, and include references to suitable prior work that address varying size of action space.
* [minor] In the Introduction first paragraph, "different maps" is mentioned without any prior context, it should be rephrased. Up to that point, readers do not know that this paper is concerned with the SMAC environment only.
* [minor] In the Figure 1 caption, MADT is mentioned without any reference. Up to that point, readers do not know what is MADT.
* [minor] In the Introduction page 2, the paper writes "MaskMA is the first approach that achieves strong zero-shot capability for multi-agent decision-making". It is not clear what "strong" means, make it more precise.
* [minor] In Section 4.4 "Mask-based Training Strategy", there is a mention of "learn permanent representations". Rephrase this to be more precise about what "permanent" means and provide evidence for this "permanence", or rephrase the text.

**Strengths And Weaknesses:**

Strengths:
1. It is good to see that two simple methods lead to significant improvement in results over baseline
2. Thorough and strong empirical results back up all the claims.
3. This paper pushes forward the study of transformer-based policies for decision-making.

Weaknesses:
1. The "Generalizable Action Representation" that is proposed to help support a varying action space size is claimed as a new contribution, but it has appeared in past work where the action space changes, e.g. [1]
2. In the formulation Section 3.1.1, the global state is defined by the set of all agent states. This formulation excludes environments that have agent-independent global state components.
2. The model architecture is not completely described, so reproducing the model may be difficult.
3. The paper acknowledges the limitation of the assumption that expert data is available, but does not discuss how this can be overcome.


[1] Yang et al. "Reinforcement learning for adaptive mesh refinement." International Conference on Artificial Intelligence and Statistics. PMLR, 2023.

---

> ### Author Response · Authors · 2024-07-09
> **Response to Reviewer rbcy**
>
> Thank you for your valuable comments and thorough suggestions. We have addressed all identified weaknesses and revised the manuscript, highlighting the modifications in blue according to the required changes. Please review the updated manuscript for detailed changes.
>
> ### Weakness
>
> > W1. The "Generalizable Action Representation" that is proposed to help support a varying action space size is claimed as a new contribution, but it has appeared in past work where the action space changes, e.g. [1]
> >
>
> Thank you for providing this reference. [1] is an impressive work that formalized **adaptive mesh refinement** (AMR) as a Markov decision process  and demonstrated for the first time that RL can outperform widely-used estimators. We recognize the similarity in the concept of handling changing action spaces. However,
>
> (1) Field of Application: GAR aims to support varying action space in multi-agent RL filed while [1] handles variable action spaces within AMR filed. The difference in application fields led to the initial miss of [1] from our literature review.
>
> (2) Implementation Details: [1] proposes three networks to handle variable-size state and action spaces and the most similar architecture to our method is graph network policy. However, the implementation details differs. Specifically, we employ direct embedding of actions (interactions between agents) while [1] utilizes the embedding of each element using graph network.
>
> (3)  Revision for Clarity:
>
> We have made the following changes and cited [1] to address your concerns in the revision:
>
> - We have revised the delivery in our manuscript **from "propose" to "introduce"** to clarify that we are adapting existing techniques to the multi-agent environment context, rather than claiming the initial invention of these methods.
> - Additionally, we have included RL for Adaptive Mesh Refinement methods [1] **in the related work**.
> - We also emphasize that our GAR method is similar to the methods in [1] in the GAR method section.
>
> These changes ensure the manuscript accurately reflects the development of our approach and acknowledges prior foundational work.
>
> ------
>
> > W2. In the formulation Section 3.1.1, the global state is defined by the set of all agent states. This formulation excludes environments that have agent-independent global state components.
> >
>
> Thank you for your detailed feedback. Our formulation in Section 3.1.1 already includes environments that have agent-independent global state components. In such cases, we can include the part of the global state visible to the agent at the current timestep as part of the agent's state for that timestep.
>
> ------
>
> > W3. The model architecture is not completely described
> >
>
> Thank you very much for your suggestion. We have added a PyTorch-style pseudocode of the model architecture and GAR in Appendix A.4 in the revision.
>
> ------
>
> > W4. The paper acknowledges the limitation of the assumption that expert data is available, but does not discuss how this can be overcome.
> >
>
> Thank you for this great suggestion. A promising approach to make use of low-quality demonstration datasets is to include the 'return-to-go' embedding, as used in decision transformers [2], in our model's input.
>
> ------
>
> ### Requested Changes
>
> > [critical] Revise statements that imply the novelty of the Generalizable Action Representation, and include references to suitable prior work that address varying size of action space.
> >
>
> Please see W1.
>
> > [minor] In the Introduction first paragraph, "different maps" is mentioned without any prior context, it should be rephrased. Up to that point, readers do not know that this paper is concerned with the SMAC environment only.
> >
>
> > [minor] In the Figure 1 caption, MADT is mentioned without any reference. Up to that point, readers do not know what is MADT.
> >
>
> > [minor] In the Introduction page 2, the paper writes "MaskMA is the first approach that achieves strong zero-shot capability for multi-agent decision-making". It is not clear what "strong" means, make it more precise.
> >
>
> > [minor] In Section 4.4 "Mask-based Training Strategy", there is a mention of "learn permanent representations". Rephrase this to be more precise about what "permanent" means and provide evidence for this "permanence", or rephrase the text.
> >
>
> Thank you very much for your meticulous review of our manuscript. We have addressed all the requested changes in the revised version of the manuscript.
>
> ------
>
> Thank you again for your valuable suggestions and for helping us enhance our work.
>
> [1] Yang, Jiachen, et al. "Reinforcement learning for adaptive mesh refinement." International Conference on Artificial Intelligence and Statistics. PMLR, 2023.
>
> [2] Chen, Lili, et al. "Decision transformer: Reinforcement learning via sequence modeling." *Advances in neural information processing systems* 34 (2021): 15084-15097.

---

> > ### Comment · Reviewer_rbcy · 2024-07-25
> > **Acknowledgement of author's response**
> >
> > I appreciate the author's detailed response to each comment, especially clarifying the global information contained within each agent's state and providing model details in pseudocode. Regarding GAR, I just meant that the description in Section 3.3 of using a variable-sized array $\lbrace FC(h^t_{i,j}) \rbrace^N_{j=1}$ of logits looks similar to the general strategy in [1], which also applies a softmax over a variable-sized array of scalar outputs produced by some shared model, regardless of whether it is a FCN or graph net. Updated claims and evidence in original review.

---

> > > ### Author Response · Authors · 2024-07-26
> > > **Acknowledgement of reviewer's response**
> > >
> > > Thank you for your updated reviews and for helping us enhance our work. We agree that GAR "applies a softmax over a variable-sized array of scalar outputs produced by some shared model," which is similar to the strategy in [1]. Therefore, we included RL for Adaptive Mesh Refinement methods [1] in the related work and emphasized in the GAR method section that our GAR method is similar to the methods described in [1].

---

### Decision · Action_Editor_yJhQ · 2024-08-12

**Recommendation:** Accept as is

**Comment:**

Initial concerns were all addressed during the rebuttal phase.

**Audience:**

The reviewers and myself are in unanimous agreement that this work tackles an important problem in the field of multi-agent systems, and should be relevant to a broad swath of TMLR's audience.

**Claims And Evidence:**

All reviewers agreed that the claims this paper made were supported empirically. There were initially some concerns around replicability since aspects of their methods (e.g. GAR) weren't sufficiently detailed, but this was solved via rebuttal. There's always a bit of uncertainty about what is truly "zero-shot" (e.g. how far OOD), but both the reviewers and myself are satisfied with the arguments and evidence provided on this front.